# A Bottom-Up and Top-Down Participatory Approach to Planning and Designing Local Urban Development: Evidence from an Urban University Center

**Teodoro Semeraro[1,*], Zaccarelli Nicola[2], Alejandro Lara[3], Francesco Sergi Cucinelli[4] and Roberta Aretano[5]**

[1] University of Salento, Department of Biological and Environmental Sciences and Technologies, Ecotekne, Prov. le Lecce Monteroni, 73100 Lecce, Italy
[2] European Commission – DG-Joint Research Centre, Directorate for Energy, Transport and Climate, Westerduinweg, 3, NL-1755 LE Petten, The Netherlands; nicola.zaccarelli@gmail.com
[3] University of Concepción Department of architecture Victor Lamas 1290, Chile; alejandrolara@udec.cl
[4] Francesco Sergi Cucinelli, Intercultural linguistic Mediator: Via Cavalielli Dell'ordine di Vittorio Veneto, 43, 73100. Lecce, Italy; fraseku@gmail.com
[5] Environmental consultant, Via San Leucio 15, 72100 Brindisi, Italy; roberta.aretano@gmail.com
**\*** Correspondence: teodoro.semeraro@unisalento.it Tel.: +39-320-8778174

**Abstract:** The urban area is characterized by different urban ecosystems that interact with different institutional levels, including different stakeholders and decision-makers, such as public administrations and governments. This can create many institutional conflicts in planning and designing the urban space. It would arguably be ideal for an urban area to be planned like a socio-ecological system where the urban ecosystem and institutional levels interact with each other in a multi-scale analysis. This work embraces a planning process that aims at being applied to a multi-institutional level approach that is able to match different visions and stakeholders' needs, combining bottom-up and top-down participation approaches. At the urban scale, the use of this approach is sometimes criticized because it appears to increase conflicts between the different stakeholders. Starting from a case study in the Municipality of Lecce, South Italy, we apply a top-down and bottom-up participation approach to overcome conflicts at the institutional levels in the use of the urban space in the Plan of the Urban University Center. The bottom-up participation action analyzes the vision of people that frequent the urban context. After that, we share this vision in direct comparison with decision-makers to develop the planning and design solutions. The final result is a draft of the hypothetical Plan of the Urban University Center. In this way, the bottom-up and top-down approaches are useful to match the need of the community that uses the area with the vision of urban space development of decision-makers, reducing the conflicts that can arise between different institutional levels. In this study, it also emerges that the urban question is not green areas vs. new buildings, but it is important to focus on the social use of the space to develop human well-being. With the right transition of information and knowledge between different institutional levels, the bottom-up and top-down approaches help develop an operative effective transdisciplinary urban plan and design. Therefore, public participation with bottom-up and top-down approaches is not a tool to obtain maximum consensus, but mainly a moment of confrontation to better address social issues in urban planning and design.

**Keywords:** urban planning; urban space; urban regeneration; planning process; public participation

## 1. Introduction

Urban areas are ecosystems characterized by natural and artificial elements such as buildings, roofs, underground pipes, and green areas that are mainly related to human well-being. The urban ecosystem is not self-regulating but is "regulated" by humans [1–3]. The urban area is, therefore, a complex socio-ecological system where various communities can overlap and interact to a greater or lesser extent and co-evolve with their environment through change, instability, and mutual adaptation [2,4]. Therefore, the evolution of an urban ecosystem is influenced by social decisions or needs and by stakeholders' heterogeneity (for instance, culture, education, religion, vision, interest) [5–8]. These institutional levels include decision-makers like administrative and public institutions that plan the socio-ecological system at different scales: "ecosystems in urban areas", "urban areas within ecosystems", and "urban areas within regional/global ecosystems". At different scales, the boundaries of the urban ecosystem are not always well defined or clear, and therefore, boundaries of a survey area are defined considering the topic and interactions to be analyzed and on practical considerations. [5,9]. Institutional levels are hierarchical and consist of vertical relations between actors of the top and bottom of the levels. Therefore, the urban area has to be planned like a socio-ecological system where the urban ecosystems and institutional levels interact with each other in a multi-scale analysis. The urban planning and design have to create a synergy between different institutional levels (Figure 1) [10–13].

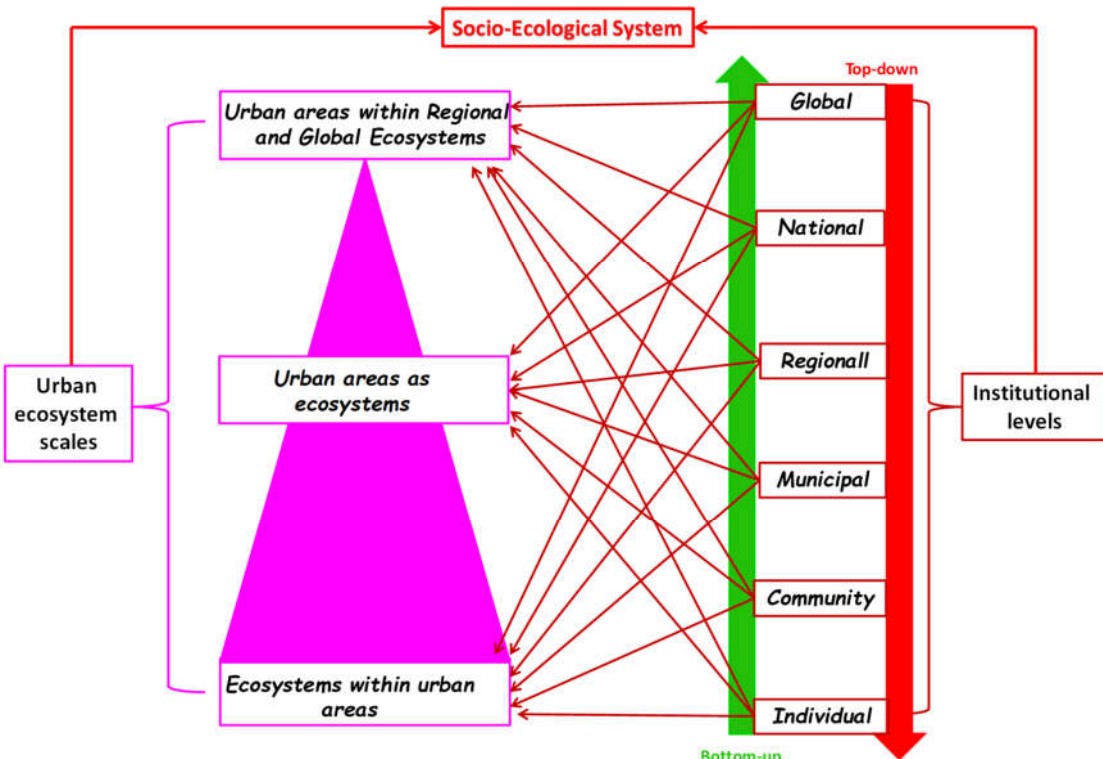

**Figure 1.** Schematic representation of the relationship between urban ecosystem scales and institutional levels in the socio-ecological system [5,6,9].

In many industrialized cities, urban planning must address the phenomenon of "shrinking cities" [14]. These cities have experienced a significant de-urbanization linked to the loss of functionality of some urban areas or buildings due to the decline of the manufacturing industry, migration, and depopulation [15,16]. Consequently, urban areas are characterized by free or temporarily not used urban spaces as a result of technological, economic, and social evolution. In many cases, these urban areas could be brownfield sites: "streets with vacant storefronts, underutilized social and technical infrastructures, and neglected parks and squares" [16,17].

Identifying new functions in urban spaces—either built or otherwise—in a transitioning economy and society is the main focus of resilient thinking, which has to recognize the complex and non-linear dynamic of economic and socio-ecological interactions [18–20].

Currently, the main urban planning and design use the top-down approach, where planners are considered "the experts" who put forward the proposal and then share it with others, mainly the decision-makers that can approve or reject the urban plan [13,21–24]. This generates stakeholders' conflicts in the type of use of the urban space, environmental protection, the interest of residents, labor conditions, economic development, and the identities of urban areas [8,25,26].

The planning of the urban space has to be considered a "public affair", aiming to envisage the right use of urban spaces considering the socio-ecological and cultural context of reference and solving conflicts in the choices or preferences in the use of destination of the urban space between stakeholder groups. Being able to evaluate the "awareness, value judgments, behavior, and attitudes" of the citizen in relation to urban space is an important task for a successful plan of urban transformations [27–30]. To create a social and shared vision of possible scenarios that can transform the territory, a prominent role must be given to stakeholders' needs, opinions, and interests, but also fears and doubts, in order to include their vision in the development of the urban space that they use [31–33].

Therefore, urban planning needs to combine bottom-up and top-down approaches, including stakeholder's participation with strategic spatial planning at different urban levels [34]. Public participation helps to understand the aspirations of stakeholders on possible urban development. Moreover, perception stimulates different stakeholders to develop ideas and proposals based on their knowledge, attitudes, and habits, providing greater awareness of their role in urban development. It is an action in urban design useful for increasing the ability to make effective planning choices [30]. For this reason, scholars consider stakeholder participation as one of the main aspects to take into consideration in order to guarantee the quality of urban planning [35,36].

On the one hand, there are many examples of bottom-up and top-down approaches in policy activities that are mainly focused on the management of natural resources or services (e.g., energy policy, climate change, watershed management, mobility, agricultural, environmental) on municipal, regional, national, and international programs. On the other hand, these approaches are less frequently used on small urban land use planning and design [37–39]. Although these approaches have attempted to include community stakeholders, this has often proved problematic, and planning guidelines do not yet consider design principles that foster social learning, knowledge exchange, and power-sharing [7,40–42]. Mainly, public participation may not always yield a mutually acceptable solution, especially when the interests of stakeholders are diverse and conflicting [38]. Often, top-down and bottom-up urban planning approaches are sometimes considered incompatible because they can produce conflict and fragmentation in the built new environment vision between different urban levels and stakeholders [13,43,44].

This work wants to develop a planning-process of the urban space transformation able to create feedback between different stakeholders at different institutional scales to reduce the mismatch between governance levels and the scales at which people benefit from urban space: from the need of the single individual to the development vision of the decision-makers [5,6,45]. Starting from a practical case study, we propose a combination of a bottom-up and top-down methodology capable of developing a participated urban plan, harmonizing the various stakeholders' interests that act at the different administrative levels and integrating ecological and socio-economic components in the context in which it is inserted [31,45].

Mainly, the study is focused on the Plan of the Urban University Center (PUUC), involving the creation of new university lecture halls in a university urban space that was the research site of tobacco production. Considering that the University represents the main stakeholder, as it is the owner of the urban space, the planners tried to satisfy university needs with the urban transformation vision of the decision-makers (top-down), also taking into account the aspirations of the citizens that act in the context area of the PUUC (bottom-up).

We hope to identify the best solutions for the use of urban spaces to integrate the citizens' visions with those of the planners and of the different public institutions that have an administrative role in choosing the final destination of the area.

## 2. Materials and Methods

### 2.1. Study Area

The study area is a free space of the university center in the Municipality of Lecce, Apulian region, South of Italy (Figure 2). The presence of the university has greatly influenced the economy of the district, favoring the opening of numerous commercial activities, such as bars, take-aways, restaurants, bookshops, and pubs. Furthermore, the real estate business linked to the rental or sale of student apartments has benefited from the situation. From the cultural and social point of view, the free urban space is located within the former Agricultural Research Center (ex CRA), which was used in the past for tobacco research activities. Currently, the ex CRA is employed for university lectures. The urban free space of interest is characterized by herbaceous vegetation with no ecological value (Figure 2B).

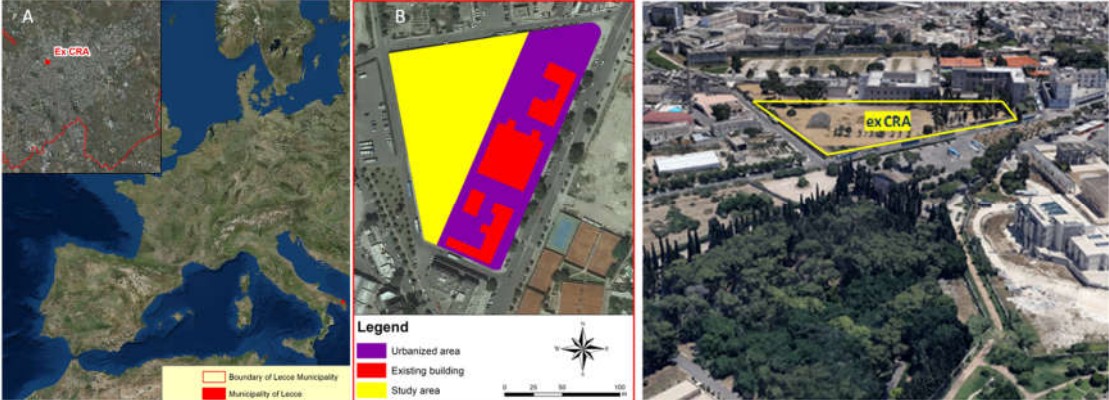

**Figure 2.** (**A**) Municipality of Lecce and location of the study area; (**B**) study area with reference to the context of the former Agricultural Research Center (ex CRA).

Near the ex CRA, there is a large green urban park, the cemetery of Lecce (classified as a historical asset for its architecture), and the Monastery of the Olivetans (founded at the end of the XII century, and currently used by the Department of Historical Studies of the University). To the west of the ex CRA, there is a main road, the Castle Charles V, and the ancient city walls. The north and south parts of the ex CRA have no significant neighboring elements.

### 2.2. Focus of the Planning Question

The Italian Inter-Ministerial Committee for the Economic Planning (CIPE) has identified and allocated resources in favor of interventions of strategic national and regional importance for the implementation of the national plan for the South's strategic priority: "Innovation, research, and competitiveness". One of the projects included in the plan for the South is located within the "Urban University Center of the ex CRA". Mainly, in an area of about 11,186 square meters, the university developed an urban plan involving the construction of a new building of about 3100 square meters for educational activities, a parking area of 1734 square meters, and a recreational green area of 6322 square meters. The new area will be realized near other university buildings, and together they will form a widespread urban university campus. The budget for the execution of the plan is EUR 8,000,000.

During the first phase of the planning activity, some regional authorities expressed a favorable opinion of the new PUUC because the plan did not cause negative environmental impact. However, the Ministry of Cultural Heritage, through the Superintendence, was the main institution that

opposed the university urban plan. The Superintendence considers that the PUUC will alter the harmony between existing buildings and the identity of the area, and hence it only promotes the development of a green lung.

Considering the institutional conflict, this research wanted to develop bottom-up and top-down participation processes approach able to orient future use of free urban spaces.

### 2.3. Design Approach

This study was developed considering bottom-up and top-down approaches in the socio-ecological system [46,47]. Mainly, the stakeholders' participation process was designed considering different roles in the transformation of the socio-ecological system. The designers of the University and Superintendence are considered decision-makers that can directly choose the typology of the transformations (top-down). The citizens are considered as users of the urban space on which the choices of the decision-makers are reflected. However, at the same time, the citizens can revolt against the choice of decision-makers and condition the final result (bottom-up).

The work is organized as follows (Figure 3):

Stage 1 develops a method to get information on citizens' perceptions, considering social issues linked with urban space;

Stages 2 and 3 are dedicated to obtaining data and analyzing the results;

Stage 4 develops the discussion of results of the citizens' participation;

Stage 5 discusses the results of the questionnaires with the decision-makers.

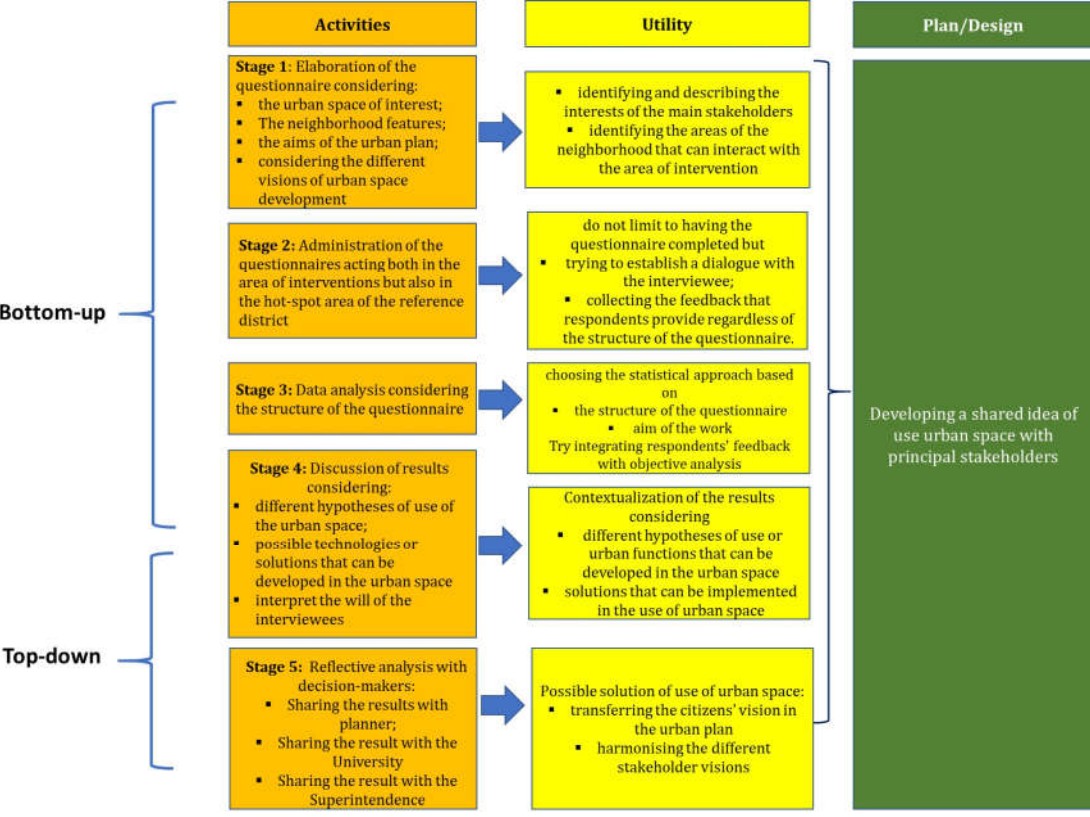

**Figure 3.** The conceptual work models that we have developed for the top-down and bottom-up approaches.

### 2.4. Bottom-Up Activity: Questionnaire Survey

The study was planned by taking into consideration the microscale, which encompasses the urban space, the structure of buildings, the relationship between them, and their interaction with other elements of the neighborhoods [48].

The work used face to face questionnaires to gather information about what citizens "feel, hope, wish, approve, or disapprove" for the future use or transformation of the identified urban space [49,50].

The survey was developed so as to include different types of citizens and users, such as students and people who live or work near the area where the project should be developed. This is important in order to explore their opinions and preferences on the possible uses of the free space of the ex CRA. Questionnaires were administered from 15 April 15 2018 to 30 May 2018, both in the morning and in the afternoon during working days and holidays. This was necessary to better characterize the typology of individuals who frequent the area of interest.

The questionnaires were delivered in three different places: the "ex CRA" area, the parking lot in front of the area of interest, and in the urban park "Belloluogo" (Figure 4), as these are the principal hotspot areas of the neighborhood in the context of the urban space of interest.

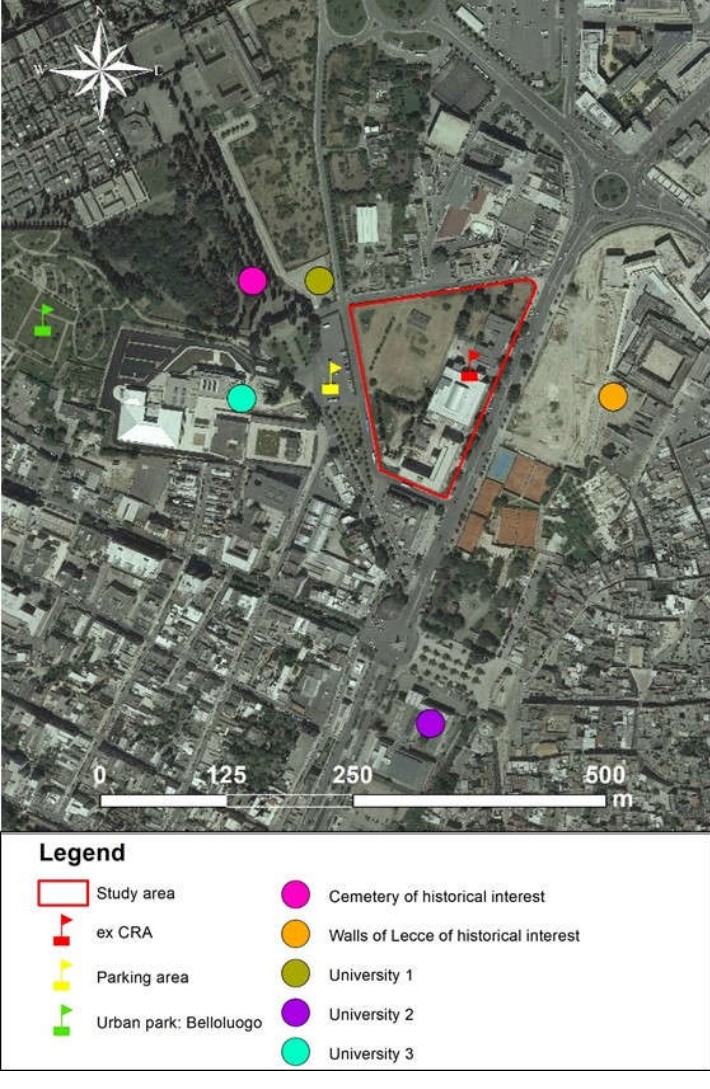

**Figure 4.** Localization of the points of administration of questionnaires, historical elements, and urban university sites.

We used Sierra's formula to detect the sample size of the number of individuals to interview [50]:

$$\text{Sample Size} = \frac{4 * N * p * (1 - p)}{E^2(N - 1) + 4 * p * (1 - p)}$$

where $N$ is the number of inhabitants; $E$ is tolerated error; $p$ is the portion of the variable in the population" [39].

Questionnaire structure

The questionnaire was structured in order to evaluate four main dimensions: the main users of the area; the historical relevance of the place (Items 1–3), the building size and harmonious insertion in the urban context (Items 4–7), and the importance of developing green areas compared to other options (Items 8–9). In particular, considering Item 9, three alternatives in the use of the area were indicated, and respondents could express a preference value. This last aspect was developed after taking inspiration from Directive 2001/42/EC and national law (D.lgs 152/2006), which provide for the analysis of different planning hypotheses in the drafting of the Strategic Environmental Assessment (SEA) to identify the best possible solution [51,52].

The questions were formulated with a simple and clear structure to be filled out quickly by ticking the preference box. Specifically, respondents were asked to express a preference using a five-point for the second section and a ten-point for the third section. In this way, respondents indicate their level of agreement to a statement. In the present survey, the ten-point scale was used when respondents could express their preference for the different design solutions proposed (e.g., [53–56]).

The data analysis was performed with the descriptive statistics for the first seven items, while the last items were analyzed through tables of contingency. The contingency tables are used to represent and to analyze the relationships between two or more variables, through the study of their combined frequencies [57].

Before the survey, a pilot study was conducted, and five questionnaires were administered to people to verify whether the proposed items were adequate and easily understood.

### 2.5. Top-Down Activity: Sharing the Questionnaire Results with the Main Decision-Makers

As the final part of this work, we discussed the results of the questionnaires with the decision-makers, who, at that moment, had a different vision for the use of urban space. In this way, we try planning a new hypothesis of the shared urban space transformation.

This represents a simple exercise to try to design the new PUUC considering citizens' visions.

This was developed with informal appointments with decision-makers like the designers of the University and Superintendence.

In these appointments, starting from the original PUUC, we discussed potential new solutions of the PUUC and analyzed how the main results of the questionnaires could be incorporated in the new planning of the urban space. After that, we developed an illustrative conceptual graphic of one of the possible new urban space transformations. For practical reasons of this research, we have not considered the turnover between the various decision makers in time.

### 3. Results and Discussion

#### 3.1. Bottom-Up Activities

#### 3.1.1. Stakeholders Characterization

For the study area, the minimum sample size is 382 individuals. Therefore, our 624 questionnaires can be considered as representative of the population that characterizes the study area. Mainly, 42% of the questionnaires were compiled in the ex CRA area, 33% in the parking area, and 25% in the urban park "Belloluogo" (Figure 4).

Table 1 shows the main socio-demographic characteristics of the sample interviewed. It is possible to note that the main users of the area are young people and students. In particular, most of

the interviewees were students under 25. This result could be strongly influenced by the presence of two university centers.

**Table 1.** Socio-demographic characteristics of the sample.

| Characteristic | Category | % |
|---|---|---|
| Gender | Male | 35 |
| | Female | 62 |
| No answer | | 3 |
| Age | From 19 to 25 | 70 |
| | Over 25 | 30 |
| Occupation | University students | 71 |
| | Other activity | 29 |
| Residence | Municipality in the Province of Lecce | 68 |
| | Outside | 32 |

3.1.2. Historical Relevance of the Place.

Analyzing the historical relevance of the study area, it is clear how almost all of the respondents considered this area relevant to the historical and cultural point of view. Only 19% said that the area is "not very important" and only 3% that it is "not at all important". However, the surveyed people did not know the history of the ex CRA, or where precisely this building was located. Therefore, the surveyed respondents consider this building historically important for the urban context in which it is located, rather than for the specific history of the structure. The distribution of answers obtained is similar in the different sampled areas (Figure 5).

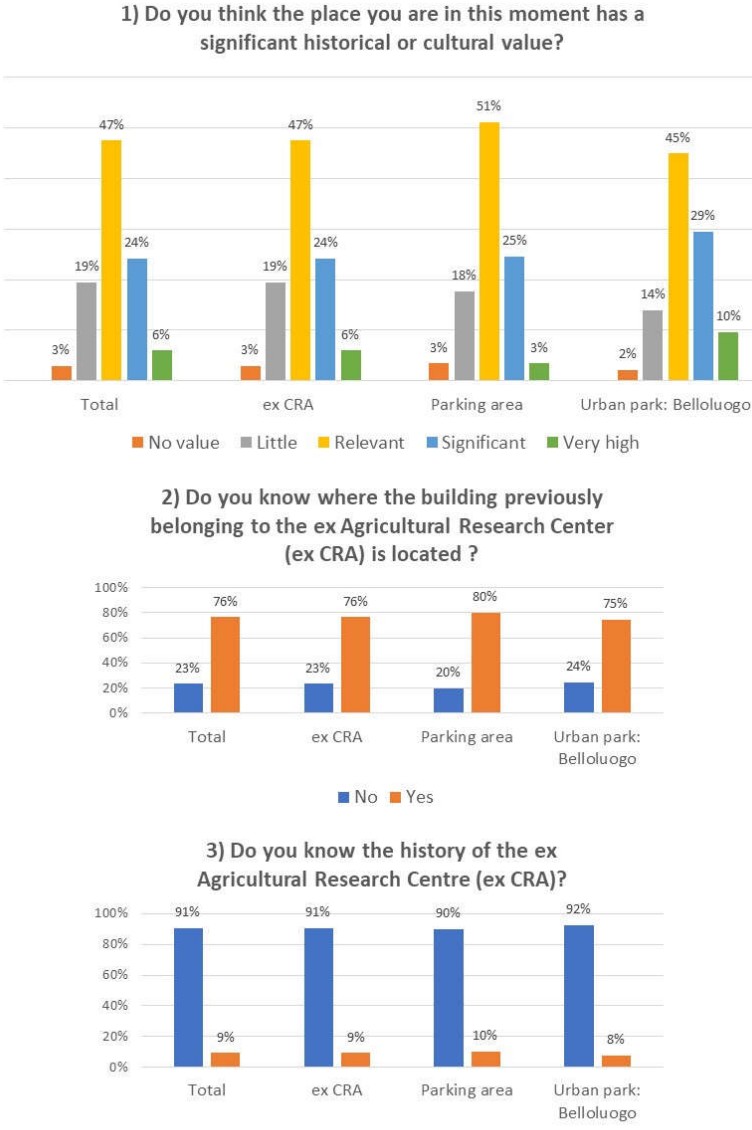

**Figure 5.** Answers to Questions 3, 4, and 5, concerning the historical relevance of the study area.

### 3.1.3. New construction in urban space

Subsequently, the opinion of the surveyed respondents was investigated with reference to the construction of a new building for teaching purposes within the former CRA. In particular, 43% of the respondents answered that the possibility of a new building altering the architectural harmony between existing buildings depends on the type of building that will be built. However, this building will not be a disturbance to the livability of the area (Figure 6).

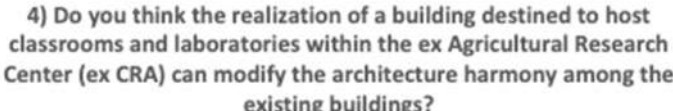

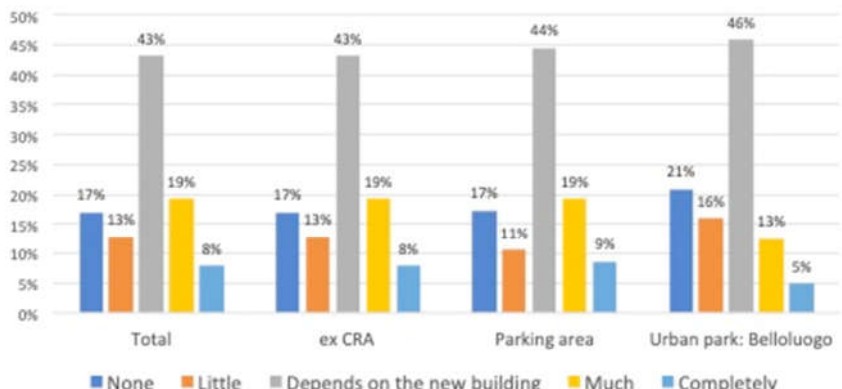

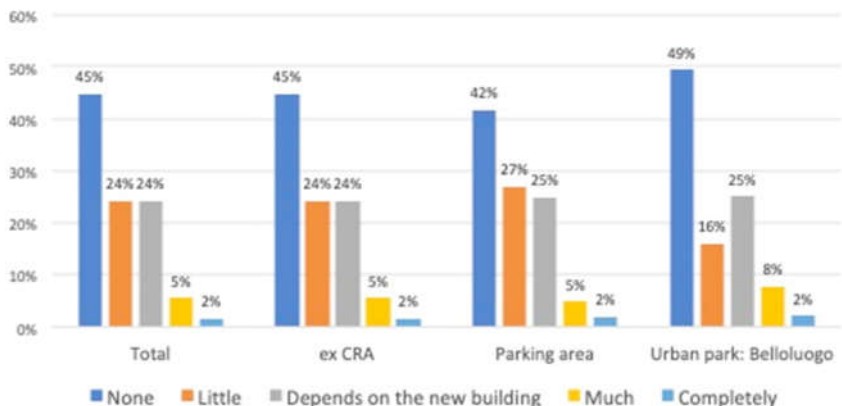

**Figure 6.** Answers to questions on the possibility of a new building inside the former CRA.

According to the surveyed respondents, the width and height of a new building could also be the same as the size of the central building existing within the former CRA, or even wider and higher. In fact, these two options were the second most selected choices in all the three sampling areas (Figure 7).

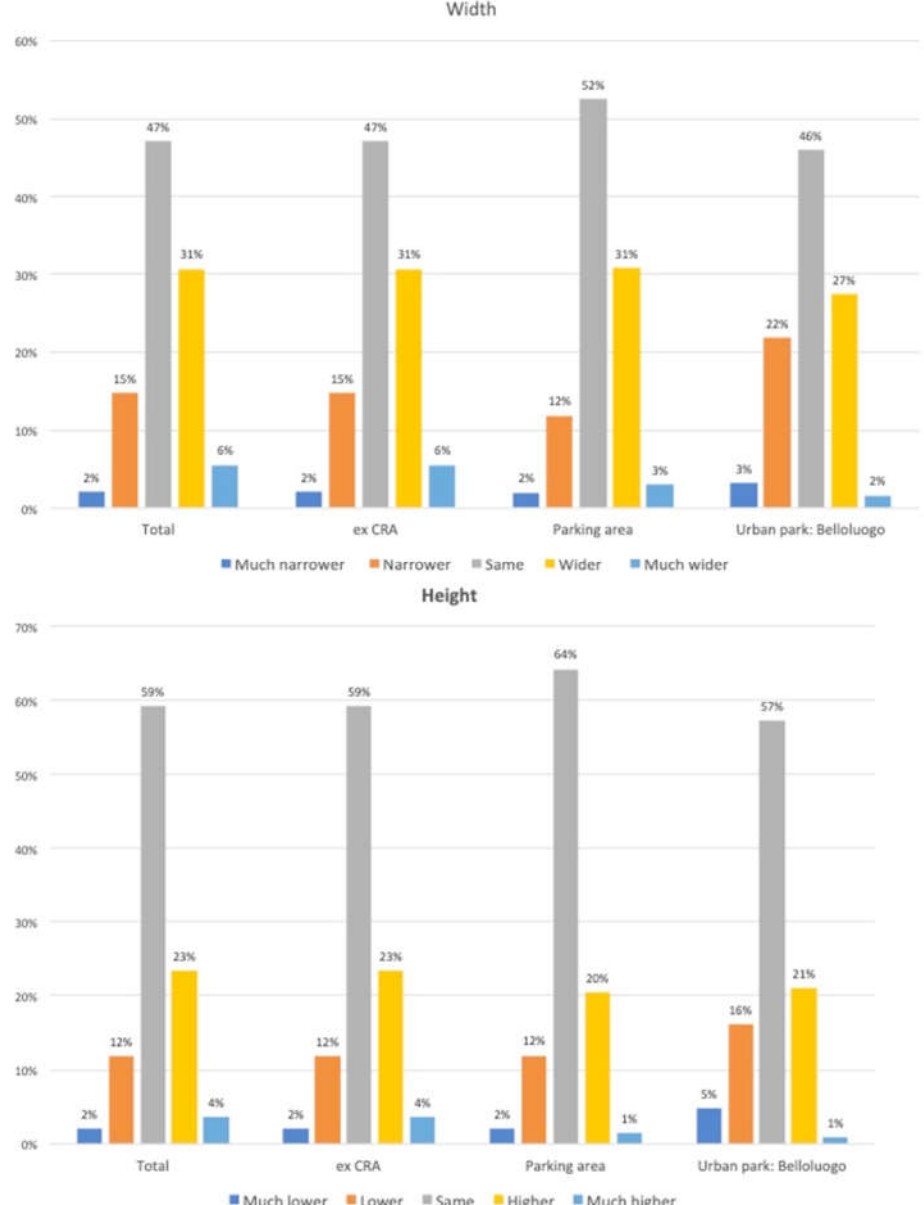

**Figure 7.** Answers to questions on the possible width and height of the new building that is to be realized within the former   CRA.

However, the construction of the new building should not lead to the elimination of the currently unused greenhouse. According to the surveyed respondents, the greenhouse should be used for social activities (Figure 8).

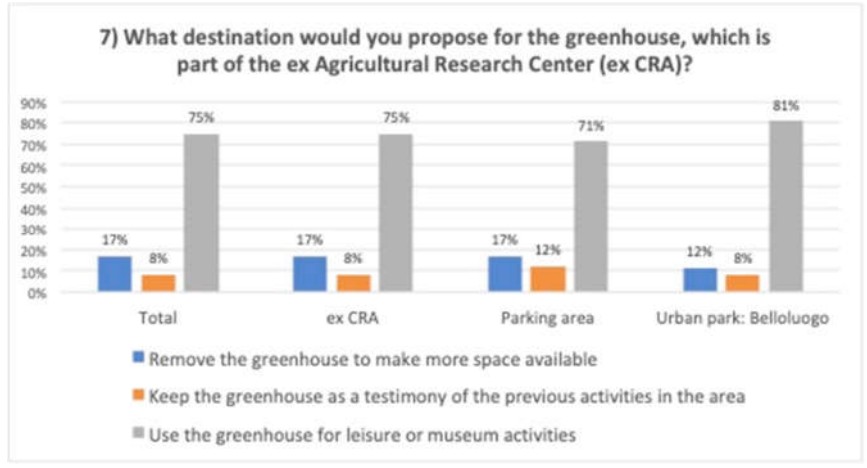

**Figure 8.** The question related to the greenhouse.

### 3.1.4. Potential actions that can be developed in the urban space

In the final questions of the questionnaire, we investigated the main actions of the plan to be launched within the former CRA (Figure 9). Surprisingly, it emerged that "improving the quality of urban green areas" is not among the main prerogatives indicated by the surveyed respondents. Overall, the main preferences were "finding a right harmony between use, green spaces, and economic activities" and "favoring the development of social activities (concerts, parties, and shows)". However, the surveyed respondents within the ex CRA expressed their preference "to create new classrooms and laboratories to enhance the university's educational offerings and accommodate a larger number of students" and "favoring the development of social activities" (Figure 9).

From the analysis of the contingency tables, statistically significant differences emerge. Mainly, there are differences between the alternatives "a", "b", "d", and "i". ($p < 0.01$). However, considering three sampling points (Figure 9), there are no statistically significant differences in the choice for the alternatives "a", "d", and "i", while there are statistically significant differences for the choice obtained for the alternative "b" in the different areas. Furthermore, the results show statistically significant differences in the choice of the solutions proposed considering the age of the surveyed respondents. The respondents aged 25 years and under expressed their main preferences as "i", "d", and "b", while those over 25 expressed as main preferences "a", "i", and "d" (Figure 9).

The numbers in the cells represent the frequencies relative to the appreciation value for each proposed alternative. The answers were analyzed, considering both the single survey areas and the whole (Figure 9a). In Figure 9b, the answers were analyzed considering two age groups: under 25 years and over 25 years.

## 8) What actions would you perform in a transformation project of the ex CRA?

a)　Enhancing the city green areas

b)　Realizing new classrooms and laboratories to enhance the educational activities and serve a higher number of students

c)　Eliminating the perimetral fencing to allow a higher connection with other elements of the area: bus parking, car parking, park Belloluogo, the Olivetani monastery, shops in the surroundings, cemetery, Carlo Pranzo area and the fence walls

d)　Finding a harmony among the fruition, the realization of green areas and the creation of economic activities

e)　Realization of footpath and bus platforms

f)　Realization of an area always accessible to the residents

g)　Restoration of the greenhouse

h)　Increasing the number of parking lots

i)　Promotion of social activities

### Table a

| Appreciation values | Total | | | | | | | | | Ex Cra | | | | | | | | |
|---|---|---|---|---|---|---|---|---|---|---|---|---|---|---|---|---|---|---|
| | a | b | c | d | e | f | g | h | i | a | b | c | d | e | f | g | h | i |
| 0 | 2 | 5 | 25 | 3 | 22 | 5 | 32 | 41 | 20 | 1 | 1 | 9 | 2 | 9 | 2 | 10 | 8 | 15 |
| 1 | 3 | 3 | 6 | 2 | 8 | 3 | 10 | 17 | 2 | 1 | 1 | 3 | 2 | 2 | 2 | 6 | 5 | 1 |
| 2 | 8 | 5 | 19 | 4 | 20 | 5 | 14 | 16 | 7 | 4 | 1 | 8 | 2 | 6 | 4 | 6 | 8 | 4 |
| 3 | 13 | 9 | 29 | 8 | 23 | 12 | 23 | 20 | 6 | 3 | 2 | 11 | 2 | 7 | 2 | 11 | 5 | 1 |
| 4 | 10 | 16 | 23 | 10 | 18 | 9 | 16 | 30 | 8 | 4 | 3 | 10 | 3 | 6 | 6 | 7 | 9 | 5 |
| 5 | 39 | 25 | 34 | 21 | 60 | 40 | 44 | 56 | 34 | 13 | 7 | 17 | 14 | 27 | 23 | 18 | 21 | 17 |
| 6 | 29 | 36 | 42 | 47 | 54 | 43 | 53 | 48 | 34 | 9 | 10 | 14 | 20 | 25 | 17 | 19 | 25 | 17 |
| 7 | 67 | 69 | 58 | 65 | 58 | 78 | 81 | 71 | 61 | 31 | 21 | 24 | 24 | 27 | 33 | 32 | 28 | 25 |
| 8 | 123 | 113 | 86 | 101 | 87 | 134 | 86 | 73 | 95 | 53 | 51 | 30 | 42 | 32 | 51 | 33 | 27 | 35 |
| 9 | 61 | 80 | 70 | 79 | 78 | 56 | 60 | 53 | 62 | 28 | 34 | 35 | 36 | 32 | 17 | 28 | 27 | 23 |
| 10 | 213 | 207 | 176 | 228 | 140 | 183 | 149 | 143 | 239 | 87 | 103 | 73 | 87 | 61 | 77 | 64 | 71 | 91 |

| Appreciation values | Parking area | | | | | | | | | Urban park: Belloluogo | | | | | | | | |
|---|---|---|---|---|---|---|---|---|---|---|---|---|---|---|---|---|---|---|
| | a | b | c | d | e | f | g | h | i | a | b | c | d | e | f | g | h | i |
| 0 | 1 | 0 | 9 | 1 | 6 | 0 | 13 | 10 | 4 | 0 | 4 | 7 | 0 | 7 | 3 | 9 | 23 | 1 |
| 1 | 2 | 0 | 1 | 0 | 2 | 1 | 3 | 3 | 1 | 0 | 2 | 2 | 0 | 4 | 0 | 1 | 9 | 0 |
| 2 | 3 | 3 | 7 | 2 | 8 | 1 | 5 | 4 | 3 | 1 | 1 | 4 | 0 | 6 | 0 | 3 | 4 | 0 |
| 3 | 7 | 4 | 12 | 5 | 10 | 10 | 8 | 8 | 3 | 3 | 3 | 6 | 1 | 6 | 0 | 4 | 7 | 2 |
| 4 | 5 | 6 | 10 | 3 | 5 | 1 | 7 | 12 | 3 | 1 | 7 | 3 | 4 | 7 | 2 | 2 | 9 | 0 |
| 5 | 16 | 5 | 11 | 5 | 19 | 12 | 20 | 25 | 9 | 10 | 13 | 6 | 2 | 14 | 5 | 6 | 10 | 8 |
| 6 | 14 | 12 | 14 | 13 | 14 | 19 | 16 | 11 | 13 | 6 | 14 | 14 | 14 | 15 | 7 | 18 | 12 | 4 |
| 7 | 23 | 29 | 23 | 24 | 18 | 22 | 33 | 26 | 22 | 13 | 19 | 11 | 17 | 13 | 23 | 16 | 17 | 14 |
| 8 | 47 | 40 | 32 | 37 | 42 | 54 | 30 | 35 | 38 | 23 | 22 | 24 | 22 | 13 | 29 | 23 | 11 | 22 |
| 9 | 18 | 35 | 22 | 26 | 29 | 21 | 20 | 15 | 24 | 15 | 11 | 13 | 17 | 17 | 18 | 12 | 11 | 15 |
| 10 | 65 | 67 | 60 | 85 | 48 | 60 | 46 | 52 | 81 | 61 | 37 | 43 | 56 | 31 | 46 | 39 | 20 | 67 |

### Table b

| Appreciation values | Age≤25 years | | | | | | | | | Age≥26 years | | | | | | | | |
|---|---|---|---|---|---|---|---|---|---|---|---|---|---|---|---|---|---|---|
| | a | b | c | d | e | f | g | h | i | a | b | c | d | e | f | g | h | i |
| 0 | 2 | 1 | 16 | 2 | 11 | 3 | 23 | 17 | 16 | 0 | 4 | 9 | 0 | 10 | 2 | 8 | 23 | 4 |
| 1 | 3 | 2 | 5 | 2 | 6 | 3 | 9 | 12 | 1 | 0 | 1 | 1 | 0 | 2 | 0 | 1 | 5 | 1 |
| 2 | 8 | 4 | 13 | 4 | 14 | 4 | 11 | 11 | 7 | 0 | 1 | 5 | 0 | 6 | 1 | 2 | 5 | 6 |
| 3 | 11 | 7 | 21 | 4 | 13 | 8 | 19 | 14 | 3 | 1 | 1 | 7 | 3 | 9 | 3 | 3 | 5 | 2 |
| 4 | 9 | 11 | 18 | 7 | 13 | 8 | 14 | 23 | 8 | 1 | 5 | 5 | 3 | 5 | 1 | 2 | 6 | 0 |
| 5 | 25 | 14 | 24 | 17 | 48 | 31 | 36 | 42 | 27 | 12 | 10 | 9 | 4 | 9 | 7 | 8 | 13 | 7 |
| 6 | 27 | 24 | 32 | 36 | 45 | 36 | 42 | 43 | 22 | 2 | 11 | 8 | 11 | 8 | 6 | 11 | 5 | 12 |
| 7 | 52 | 48 | 47 | 47 | 44 | 64 | 68 | 51 | 47 | 12 | 18 | 9 | 17 | 12 | 12 | 12 | 15 | 12 |
| 8 | 98 | 89 | 60 | 76 | 71 | 105 | 61 | 64 | 72 | 21 | 23 | 21 | 21 | 12 | 24 | 20 | 8 | 20 |
| 9 | 47 | 68 | 58 | 61 | 60 | 40 | 43 | 42 | 53 | 14 | 9 | 12 | 16 | 16 | 15 | 14 | 8 | 8 |
| 10 | 140 | 154 | 128 | 166 | 97 | 120 | 96 | 103 | 166 | 67 | 47 | 44 | 55 | 41 | 59 | 49 | 37 | 64 |

**Figure 9.** Heat map for the analysis of possible planning actions considering the different survey areas. The colors from red to green represent a gradient of preference from low preference to high preference in relation to the number reported in the cells of the table.

The last question analyzed the preference of the surveyed respondents with respect to three project alternatives. Option 2, which involves a multifunctional use of the area and the realization of a new building, prevailed over the other two alternative projects. This preference did not show differences, considering the different sampling points or the age of the surveyed (Figure 10).

The numbers in the cells represent the frequencies relative to the satisfaction value for each option. In the first table, the answers were analyzed, considering both the single survey areas and the whole. In the second table, the answers were analyzed considering two age groups: under 25 years and over 25 years.

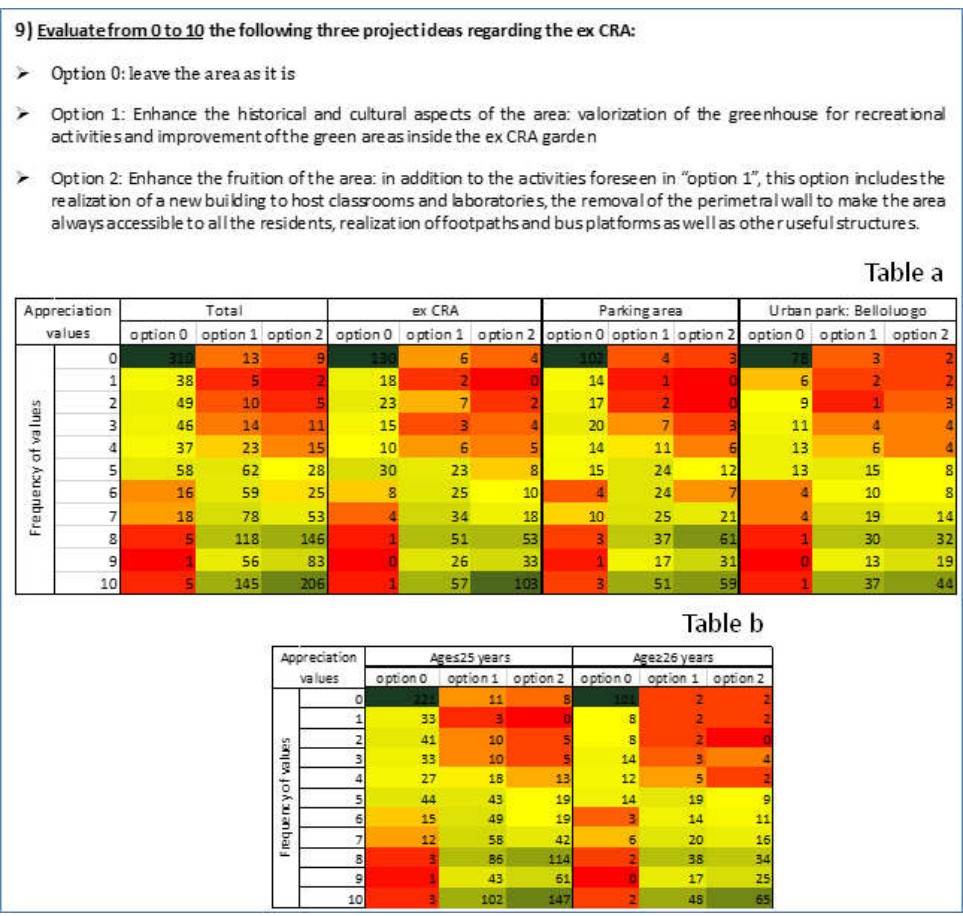

**Figure 10.** Heat map for the analysis of the results for the three project alternatives, considering the different survey areas and the age of respondents. The colors from red to green represent a gradient of preference from low preference to high preference in relation to the number reported in the cells of the table.

### 3.1.5. Relevant elements for the development of the new PUUC

The results showed that the urban space has to be planned as a social element of the context connected to ecological and economic aspects. Indeed, according to the opinions of the interviewed, in the study area, the green urban spaces were not the principal development vision of urban areas highlighted by citizens. The results showed how the main users emphasized the need to develop an integrated plan between new construction and green area that should favor economic and social development. The urban green space has to be planned as an element integrated within the development of new structures that can increase the social life of the area, including the creation of potential new buildings for economic or educative activities, creating a multifunctional center able to give vitality to the urban context of reference in different moments of the day.

From the analysis of the questionnaire, it emerged that the preservation of cultural-historical aspects should not be interpreted as prejudicial to urban transformation. What is important is to create harmony between the new urban elements that will be developed and the cultural–historical value of the area. In this perspective, according to the surveyed respondents, a new building within the former CRA should be designed as a multi-purpose center, without compromising the urban identity in which university activities, but also other social activities, can be developed, e.g., libraries, playrooms, and leisure centers.

Through participation, it was also possible to collect suggestions and specific proposals to plan the use and design of the urban space. In the initially proposed PUUC, the elimination of the greenhouse that is currently in a state of neglect was planned so as to make room for the new building.

The surveyed respondents expressed their willingness to preserve the greenhouse and use it for recreational activities (Figure 11). An interesting aspect is that many surveyed respondents were unaware of the presence of the greenhouse. Even many students who attend the area have not noticed its presence or thought that the structure was indeed a greenhouse. This probably shows how limiting the use of this area can also compromise the awareness regarding the place and its identity. Therefore, creating a green area without increasing the usability of the place or the ability of people to move within it can limit the cultural value of the area.

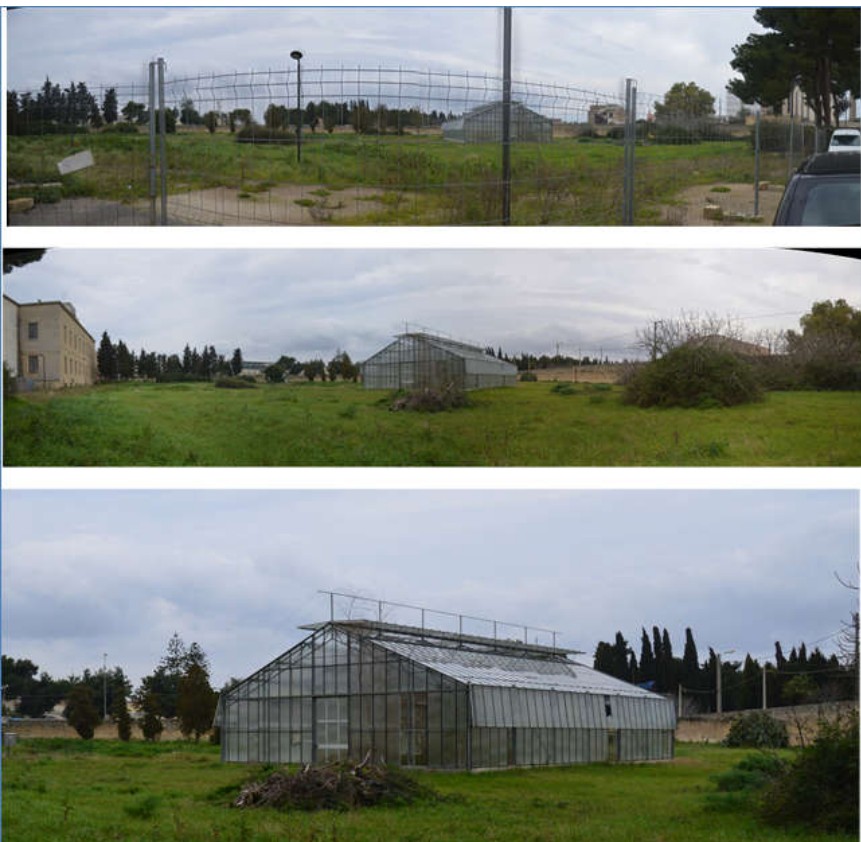

**Figure 11.** Photos of the greenhouse.

To make this area more active from a social point of view, it is fundamental to ensure greater use of the spaces and movement of individuals within the area. The possibility of an individual to perceive an area of social value and obtain benefits depends on the experience that an individual has in moving within the space, creating occasions for social interactions and enjoyable time [16,18]. The imposing enclosure wall (h = 2.50 m) that surrounds the entire area of the CRA is an obstacle, isolating this urban space from the surrounding area (Figure 12).

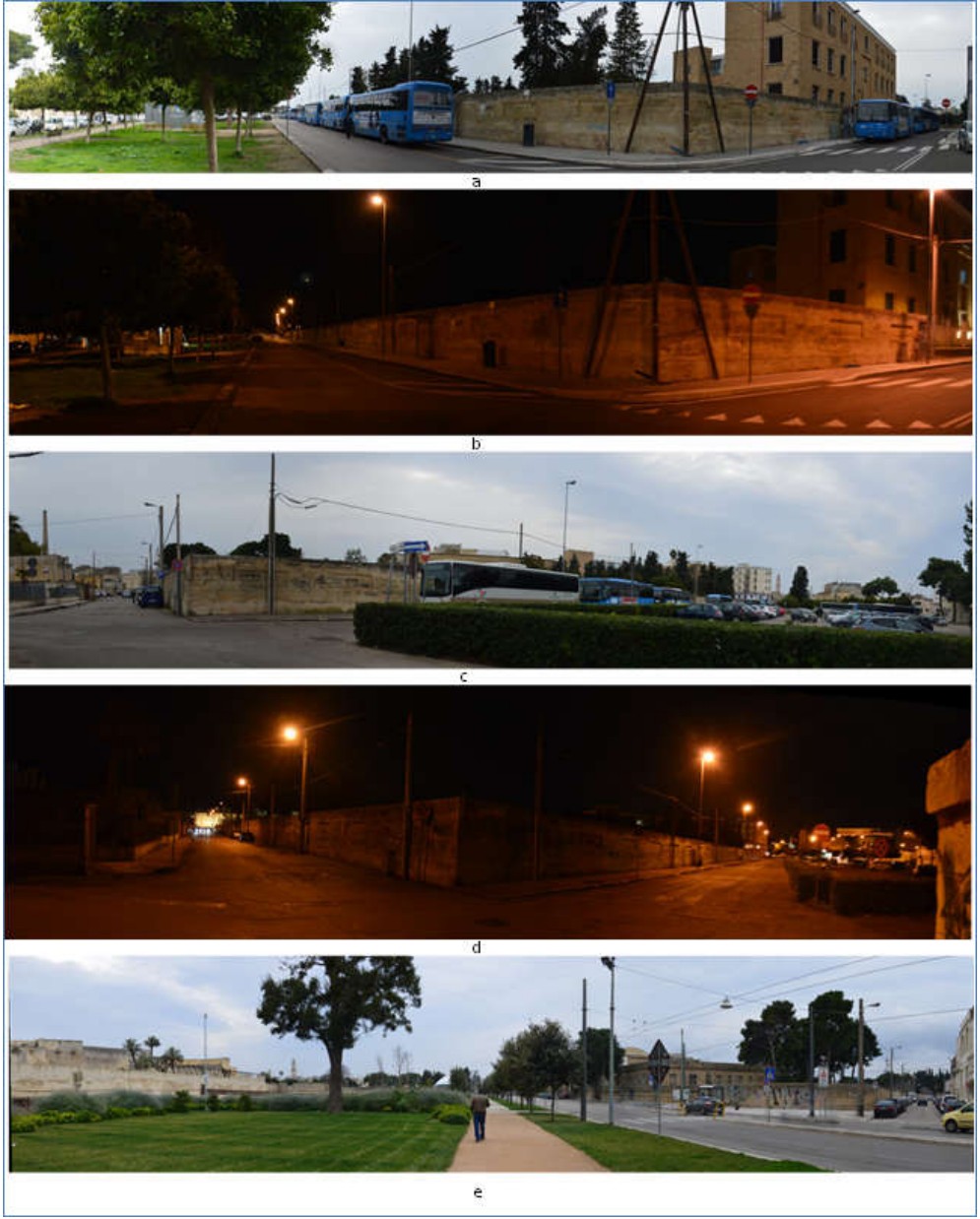

**Figure 12.** Photos of the enclosure wall. "**a**" and "**b**" are to the south–west side; "**c**" and "**d**" are to the north–west side; "e" is to the east side.

### 3.2. Top-Down Activities: Reflective Analysis Using Bottom-Up Information

This work represents an experiment of a combination of the top-down and bottom-up planning processes. In particular, in this study, questionnaire activities allowed for the identification of the interests of the main citizens who frequent the area and their social needs that will stimulate the development of the new PUUC of the ex CRA ("bottom-up"). The results of the questionnaire were shared informally with decision-makers. The results were food for thought to hypothesize the development of a plan, which, starting from different visions, could produce a shared urban transformation process. Figure 13 shows an example of possible urban visions that can emerge from the comparison between decision-makers by analyzing the results of the questionnaires (top-down). The representation in Figure 13 is a simple example of potential PUUC visions that can be developed by this process. Other design solutions can also be developed because the focus is the social

functionality that has to develop in the urban space independently of the urban elements such as buildings or other elements.

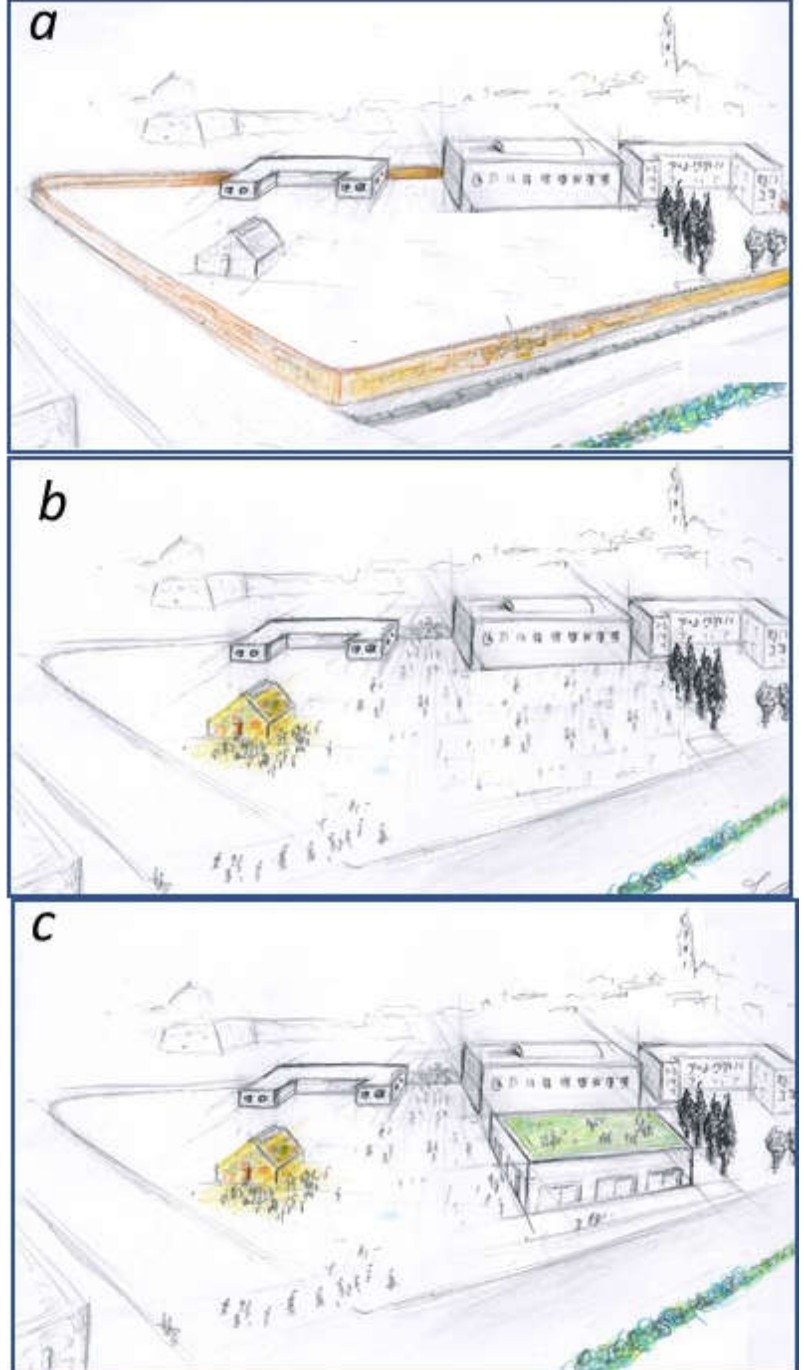

**Figure 13.** (**a**) schematic representation of the actual state of the area; (**b and c**) draft of the potential evolution of the area following and interpreting the indications of the respondents and discussion with the citizens, designers, and decision-makers. This figure represents a simple scientific exercise and does not have legal or institutional value or what this area will be in the future.

Considering these results, it emerged in these appointments that, in the new PUUC, the demolition of the wall should be further emphasized in order to encourage the development of social activities that may not necessarily be directly related to the university, but also related to leisure of a

temporary nature, and, therefore, not pre-planned. Moreover, a greater permeability of this space would facilitate the circulation of people within the urban context of reference, to better harmonize the existing valuable elements and make the buildings of the ex CRA more visible. In addition, the redefinition of the ex CRA perimeter would allow the creation of pedestrian and cycle paths—currently somewhat absent—and new bus stops supporting public transportation The urban space quality is strongly connected with the quality of public transport and the promotion of healthy patterns of walking and cycling as daily activities [58,59]. This solution would increase the type of users of the site, which is currently limited to students, to enhance the cultural and historical value of the area, now restricted within four walls. In this case, the new use of the urban space can increase the contact between citizens and urban elements that influence the cultural identity of urban space, opening the opportunity to new knowledge and new relationships between stakeholder and urban space that was limited from the no-use of urban space. The new PUUC should include the recovery of the greenhouse. The greenhouse could host indoor cultural and social activities, student associations, or a place to set up co-working. Therefore, in the new PUUC, the greenhouse will play a fundamental role in implementing the social activity of the area, increasing the fruition and experiences that the inhabitants can have within the area (Figure13b).

Another point of discussion is in regard to the possible use of the roof of a potential new building as a new social element of the PUUC. In this case, the roof can be designed as a green roof that can be used as an outdoor study area for students or small public events. Green roofs are an important tool used in residential, commercial, government, and public buildings to increase sustainability and biodiversity and decrease energy consumption, urban heat island impacts, and greenhouse gas generation in the city [60]. In this case, it helps to plan the multifunctional use of space. Therefore, green roofs can increase the quality of the buildings by reducing their impact on the urban landscape while also replacing some functions of natural areas and, therefore, assuming an important role in mental health [61] (Figure13c). However, the realization of a new building would require more discussion among decision-makers.

The solutions highlighted can make the area more dynamic, which is an important element of the human experience in urban space that incorporates both the "relationship between the person and the person-to-place relationship", improving the perception of the identity of the urban space [62].

## 4. Discussion

The main inspiration in the urban regeneration of the urban spaces or degraded areas is the realization of green areas, as they are now widely recognized and documented in guaranteeing ecosystem services useful for the wellbeing of the population [62–65]. Indeed, the type of urban space use was the main focus of the conflict between some decision-makers: implementation of the university urban center vs. green areas.

As argued by such scholars as Bourdieu, Lefebvre, and Gans [66–68], the result of the bottom-up participation showed that the urban space must be thought of as a "social space", considering the main users and producing transitions able to support good quality of life. Therefore, the challenge going forward will be to apply an increasingly advanced and nuanced understanding of urban ecology in the practice of planning and designing urban ecosystems [69,70]. Urban spaces have to be planned as dynamic areas [71], giving the opportunity to develop new structures and functions to adapt them to new social needs and economic opportunities without upsetting urban identity and ecology quality. The implementation of a university urban center vs. green area is not the main new question of the PUUC, but how this aspect can be combined to implement the social use of the area going beyond the different vision or position of the decision-makers.

In socio-ecological systems like an urban ecosystem, the bottom-up and top-down participation approaches can give both a contribution to encourage the evolution in systems and increase the resilience of the area, understood as the ability to adapt their functions and structures to social changes. In particular, the bottom-up approach allows for the identification of the main stakeholders of the area and their social needs, which, in turn, will stimulate the development of the new urban

plan of the ex CRA. This is important because it allows us to have a vision of the development of the area that is not conditioned only by the cultural background of the decision-maker, but of those who use the territory to meet the needs of everyday life. The important aspect of this approach is to actively connect the knowledge and information of bottom-up participation to decision-makers that manage the urban space at higher institutional levels.

Top-down participation, using the bottom-up information, in this case, can drive the choice and help decision-makers overcome an excessively deep-rooted view of conservation of the urban space that administratively slows down the urban regeneration process.

This would arguably help speed up the decision-making process by helping decision-makers become more aware of the transformations that are introduced in the urban context: "doing the right thing in the right place". This can be useful to produce a better acceptance of urban plans reducing the likelihood of conflicts between different experts or people that participate in the processes of planning development [31,72–74].

In this case study, participation activities started after the drafting of a first project by the university in relation to the use of the area that led the decision-makers to express a negative opinion on the project or highlighted specific critical points. From the meetings held with the decision-makers, it emerged that this approach could be useful in the initial phase of the project. However, this approach is also conditioned by the timing of urban planning. Long urban planning times can make this approach difficult to use because it can be conditioned by the turnover in management related to the decision-makers (for instance, the University, the municipality) that may require the need to restart the top-down phase. The vision of the use of urban space by current and future decision-makers can differ from previous decision-makers present when this work started.

Of course, the techniques used in this work can be improved. An important limitation of this work lies in the number of stakeholders involved in the bottom-up participation. This can affect the outcome of participation. To be representative, this method requires a great deal of effort in administering the questionnaires. Questionnaires are not the only tools to carry out the participation activity and cannot be defined as the best tool with respect to other methodologies because this depends on the scale of the investigation and type of activities. On a local level, however, questionnaires can represent valuable tools because they allow for the creation of a face-to-face relationship between planners and stakeholders. Therefore, questionnaires were used as a way to start the dialogue with the population and also to raise the interest of the interlocutors. Regarding the analysis of small transformations, it can be useful to involve citizens and their specific needs and vision in the reference context and try to put them in the final urban space planning.

Many participation approaches use the creation of thematic meetings or focus groups and tools such as online questionnaires that can open participation to a larger audience [22–24,37–40]. These actions may attract different citizens who are not necessarily users of the area, and, therefore, they may express a judgment based only on their preference or training and not on needs. This can also open a debate on the weight of the judgments expressed by those who are not familiar with or frequent the urban context of reference. We reckon that the chosen three main areas in which the questionnaires were submitted were the most functional to represent and characterize the typology of users of the urban reference ecosystem and, therefore, to analyze the social, economic, and ecological needs to be developed in the urban plan of the university (Figure 4). However, the characterization of social needs and stakeholders is the main issue of this study that needs improvement. As an alternative, using mixed methods would be an ideal solution, but this is not always feasible because of economic and time issues. For instance, in this work, the top-down activities were conducted with single appointments with the decision-makers. Therefore, in this work, we harmonized the response of decision-makers to social issues. Figure 13 is an example of the result of these activities. In the future, it can be interesting to plan the top-down participation activities by organizing focus groups between all decision-makers that participate in the authorization processes using the social issues derived from bottom-up participation as the starting point of discussion. In a similar focus group, the decision-makers should not limit themselves to

expressing an opinion based on their skills and background but should try to produce a draft project, similar to the example in Figure 13, interacting directly with each other.

Even if the questionnaire did not provide open answers where citizens could freely express their thoughts, mainly in order not to weigh down the interview, an important aspect during the compilation of the questionnaire was the dialogue established with the interviewee. Often the surveyed was not limited to the simple answers to the questions, but to an open dialogue that went beyond the structure of the questionnaire. In this way, unforeseen or planned information, consideration, and opinions were obtained.

The majority of the questionnaires were filled out by young people aged between 18 and 25 years old. The results can be influenced by the greater presence of users under 25 years and students. Probably, this expresses the main current vocation of this area. However, we want to express some "reflections" that did not emerge from the analysis of data but were derived from the dialogue with all respondents during the interviews. People from 18 to 25 years old were more interested in addressing issues concerning the development of the territory in which they live. We can state that no young people refused to complete the questionnaire, a problem that has been found in other age groups, especially men. The students were interested and encouraged to make a contribution, providing their opinion, often critical, on the issues addressed in the questionnaire to try to improve the territory in which they live and project it in the future. In addition, during the interviews, a different attitude among the interviewees emerged. Specifically, the respondents between 18 and 25 years old seemed more cooperative and willing to make a contribution to improve urban quality. Older people sounded more pessimistic about the possibility of the political class favoring actions that could produce a change and were, therefore, less constructive in providing suggestions. The questionnaires showed a different view between young people and adults (Figure 10). In the context of urban planning, those who made the final decisions are mainly adults who occupy managerial positions. Such experts would make final decisions in terms of territorial sustainability, therefore, they are thinking about future generations. However, these managers sometimes have a different vision compared to the generations that they should be protecting. For this reason, participation is an effective tool for bridging this gap.

## 5. Conclusion and Recommendation

The combination of bottom-up and top-down participation methods can be a tool through which urban planning can drive the transformation or evolution of urban spaces at different institutional levels. It can increase the interactions between citizens in a vision that "unites and inspires" to develop urban quality space helping the decision-makers to identify hypotheses of territorial development that is more suitable on the basis of present and future scenarios of economic, environmental, and social evolution.

The study shows that in the socio-ecological system, the bottom institutional levels can introduce innovation or new vision in the use of free urban space and, therefore, bottom-up participation can push or trigger the evolution of the urban ecosystem, while the top institutional level drives the change from the top-down using the bottom-up participation information in planning actions between decision-makers. Therefore, considering the adaptive cycle in social–ecological systems, the bottom-up activity can be considered as a "revolt" process that affects the "urban space", changing it from a "conservation (K)" phase, where the urban space can be kept in a state of no-social use contributing in the degradation of identities of urban area, into a "collapse or release" ($\Omega$) phase that stimulates new social uses of urban space. Therefore, the bottom-up can support the "reorganization" ($\alpha$) phase to create situations able to drive innovation, considering economic and social processes. Top-down, in this case, plays a crucial role in determining and designing the new pattern of the urban space. The system will jump into a new adaptive cycle and therefore in new urban use with new environmental, social, and economic characteristics without losing the urban identity (Figure 14).

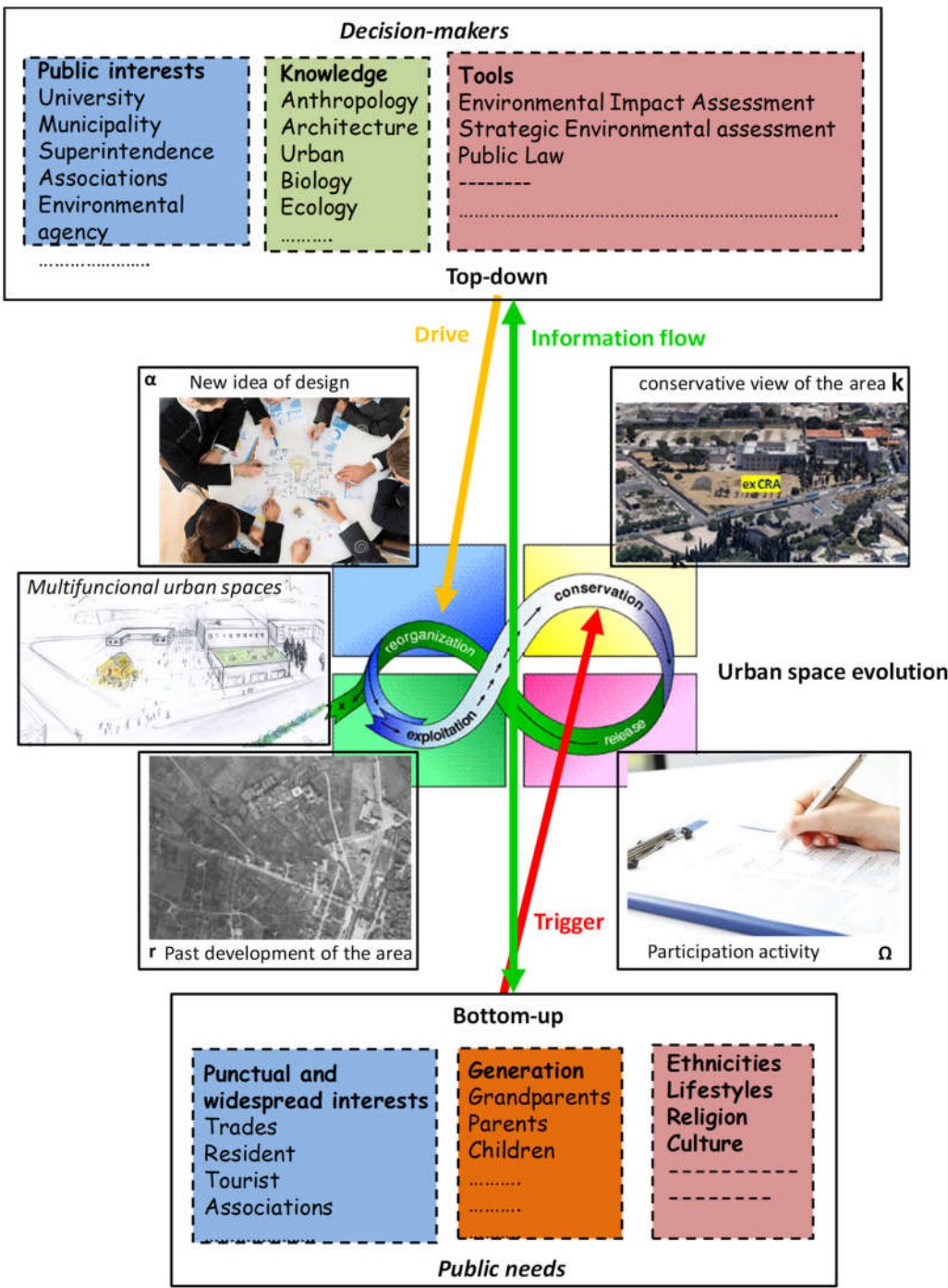

**Figure 14.** Contribution of the combination of bottom-up and top-down participation approaches in transdisciplinary planning and design of the urban space evolution considered as the socio-ecological system [10–13].

Therefore, the main aspect for the success of the bottom-up and top-down approaches is the creation of feedback between scientific knowledge derived from experiences and studies not directly connected to the characteristics of the study area and non-scientific visions deriving from those who live in the area, who express their opinions and advice based on their own life experiences. The bottom-up and top-down participation approaches can represent the base for transdisciplinary planning and design as they are useful to identify and correlate the ecological urban level and institutional levels integrating different cultural, knowledge, and generational needs, allowing the

development of a holistic vision of the evolution of the urban space (Figure 14). This is important because it allows for a vision of the development of the area that is not conditioned only by the cultural background of the decision-makers, but of those who use the territory, in an effort to meet the needs of everyday life. (Figure 14). Therefore, this approach can be useful to harmonize the differences that can emerge at different institutional levels of the urban space, going from the single individual, the community that uses the area, and the different administrative levels that make the decisions [33,34].

In this way, the participation activities were not seen as an instrument for obtaining maximum consensus, but primarily as an opportunity to take into consideration the different stakeholders' interests and to better deal with urban issues that are not yet well defined. In this paper, the bottom-up and top-down participation approaches are important to combine the need of many stakeholders (single individuals) with the vision of urban development of fewer stakeholders that take the decision (decision-makers). The effectiveness of this approach lies precisely in the ability of the decision-makers to review their own position according to the different visions without remaining in a pre-decided position. Without the flexibility of the decision-makers, this approach can fail. Therefore, the main aspect of this approach is not primarily in the techniques used, but in the ability to acquire information and knowledge and make it turn transversely, creating synergy between the various stakeholders who often act and make decisions in isolation.

This approach allows for the creation of an urban plan with more "accountability", capable of reinforcing the responsibility of choices, guaranteeing greater insurance to the citizens about the proposed transformations, and giving an account of the choices made to combat the prejudices that accompany urban transformations and making the transformation process more reliable.

An aspect of strength of this process is the possibility to analyze conflicts to start an institutional dialogue between decision-makers and final users. In this case, the Superintendence had a central role in starting a productive dialogue, changing critical issues into strong points of the urban plan.

**Author Contributions:** Conceptualization, T.S.; methodology, T.S., R.A., and N.Z., A.L.; software, T.S.; validation, N.Z., A.L., and R.A.; formal analysis, T.S. and N.Z.; investigation, T.S., A.L., and R.A.; resources, T.S. and R.A.; data curation, T.S. and N.Z.; writing—original draft preparation, T.S., R.A., and A.L.; writing—review and editing, T.S., R.A., A.L., and F.S.C.; visualization, T.S., R.A., and A.L.; supervision, F.S.C. and N.Z.; project administration, T.S. All authors have read and agreed to the published version of the manuscript.

**Funding:** "This research received no external funding

**Acknowledgments:** I thank all the guys who have given their time to this project. I thank the researcher Dr. Sara Invitto for the corrections and suggestions that she made to the paper. I also thank Dr. Francesco Dieni for the graphic representation of Figure 13. I thank the Antonie De Vitis and Niko Antonelli for the professionalism and passion with which they worked on the PUUC project.

**Conflicts of Interest:** The authors declare no conflict of interest.

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
