# Peer review of "A Bottom-Up and Top-Down Participatory Approach to Planning and Designing Local Urban Development: Evidence from an Urban University Center"

_land, doi:10.3390/land9040098_

Round 1

Reviewer 1 Report

Dear authors,

your work topic is interesting; in fact, if some reviews were implemented it could provide some guidelines for urban planning processes.

Therefore, please consider the following improvements/recommendations:

  • The discussion and conclusions should be expanded and more and references and similar studies should be added
  • Please expand the practical implications of your findings
  • There is no section regarding the study limitations and further/future research lines - should be added 
  • The English language should be revised

Besides, some reference styles are not correct: see lines 57/58.

Moreover, some language style is not expected in a scientific journal/paper: see line:490...

best,

Author Response

Response to Reviewer 1

We want to thank the reviewer 1 for the suggestions. We hope to have done the right integrations to satisfy the request increasing the value of the paper.

  • The discussion and conclusions should be expanded and more and references and similar studies should be added

Response

We implemented more references mainly in the introduction section to better describe the point of view of our study and expanded the discussion and conclusion section.

  • Please expand the practical implications of your findings

Response

Thank you for the comment. We improved the discussion and conclusion section with consideration about the practical implications of our study.

  • There is no section regarding the study limitations and further/future research lines - should be added

Response

We improved the discussion and conclusion section considering also limitations and advantages of the approach used and indication about how we want to implement this methodology in future research. We reshaped totally the conclusion.

  • The English language should be revised

Response

Done

  • Moreover, some language style is not expected in a scientific journal/paper: see line:490

Done

Thank you for the valuable feedback on our manuscript. We appreciate the chance to revise and improve our work. The comments were all addressed in the updated manuscript and point-by-point below. We found the comments very helpful and believe that now the manuscript has been improved by taking them into account. 

Reviewer 2 Report

Background: My understanding of this manuscript, having read it thrice, is that it reports a planning and designing process that combined bottom-up and top-down participatory approaches in the development of an Urban University Centre. If my understanding of the manuscript is right (and I leave the authors to confirm this), then the manuscript is interesting, but it will be difficult to grasp it entirely due to how the authors phrase their words. That aside, the paper will present some usable knowledge in the urban planning and designing field or community development. For this to happen, the authors still need to put in extensive effort to improve this manuscript as I find the manuscript interesting because it has its positives – useful visualisation techniques, and rigorous efforts put in describing what is usually a complicated process. I have provided some feedback to the authors to enable them to improve this manuscript (assuming they want to go ahead with its publication in this journal).

Title: the title of the paper, “Adopting a bottom-up and top-down participatory approach to plan and design local urban development” is simply a statement that is expressive of nothing interesting. It would have been more interesting and make for a good title if it was possible to get an idea of a unique reason for “Adopting a bottom-up and top-down participatory approach to plan and design local urban development”. Either “a bottom-up and top-down participatory approach” (or both) must be adopted “to plan and design local urban development” procedures. The title does not present a novel issue. An example (and this is based on my understanding of the paper) of a good title would have been something like “Combining bottom-up and top-down participatory approaches: Applications in the plan and design of an Urban University Centre”. With such a title, readers can easily see the innovation (how bottom-up and top-down approaches were combined) and see the application (planning and designing) and the object or case under investigation (Urban University Centre).

Abstract: The abstract does not directly relate to the reader precisely what the article is all about. It stresses the bottom-up and top-down processes carried out but do not address questions related to who carried out the processes, for what it was done, and why the approaches were adopted. I believe in the mind of the authors they believe they must have answered these questions, but their wording and phrasing of their narratives fail to carry the message across to me as a reader. For instance, in lines 21-22, the authors in expressing their objective (or critical motive of the research) stated that “This work intends to try to apply a planning process of urban spaces creating feedback between different stakeholders: community, planners,  and decision-makers.” However, this does not relate any specific message.  Why not simply put it this way, “this work embraces a planning process that…”. The later is in the present while the former is in the future. It would have been more understandable if the manuscript was written mostly in the present (or combined with past where necessary) as it is reporting what has been done (in the past) in the present. This is a minor issue but needs to be addressed as it gets in the way of comprehending the article. Due to this kind of sentence wording (which are mostly not wrong) that are awkward, it is difficult to grasp the abstract fully. This abstract needs rewriting. The authors can improve the abstract by simply adapting specific sentences that reflect or answer the following questions: What is the background of the study? Why is the study necessary now? And why is the study being reported? How was the study done? What result or outcome did the study produce? What can we conclude from the outcome of the study? And how does it add to existing knowledge? If the authors can create one or two short and straightforward sentences that answer these questions, then the abstract will project-specific messages that touch on the relevance, objective, methodology, results and conclusion of the research.

Introduction: The introduction needs to provide a more robust background of the study, as well as justify the study with literature evidence. This is especially important since the authors did not introduce a literature-based section before the “materials and Methods” section. I will not go as far as suggesting that the authors include a literature review section. Still, for them to avoid this, they need to produce a robust and expanded introduction that justifies what has been done in a related field and what the missing pieces (or gaps) exist, and why this current study fills some gap or gaps going forward. These arguments should be backed with reliable literature.

Materials & Methods (Methodology): To have a better-structured manuscript, I will advise that the “study area” (section 2) be integrated with the “Materials & Methods” (section 3). What this will mean is that the “study area” is made a sub-section (preferably the first section) in the “Materials & Methods”. This will allow for directly putting the methodology into a case study context. In the current structure, the two are isolated from each other. More so, a case study is part of a methodology because it is the object under investigation. Without the case study, the entire research will lose context and scope,  become scientifically meaningless. This is why I advise the authors to merge those two sections. Also, I think the authors should focus on describing the “study area” (that is, the Municipality of Lecce and location of the study area) in the “study area” sub-section, rather than describing the planning process. This should be followed by the “study design” sub-section, and….

Result & Discussion: The critical question which the research evoked but failed to answer is the question of, what is unique in the combination of “bottom-up” and “top-down” approaches as employed in the PUUC development? How different would the outcome have been if the two were not combined in the manner stated in the study? The answers to these questions are not easily decipherable from the study. Could it be that the answers are tucked away or hidden somewhere within the text? Maybe. But the authors can boldly state this by dedicating headings or sub-headings to make them visible in the discussion section. Another minor issue: What did the authors mean by putting lines 425-441 in italics? I didn’t get it?

Conclusion: As the discussion failed to clearly identify the novelty of the study, it is difficult for me to assess the conclusion. 

Very important: I want the authors to understand that my review has been critically constructive because their manuscript has some merit for publication. Their highly qualitative oriented methods fit a work of this sort, which needs the power of narration to argue its novelty. However, the novelty, in this case, is not yet clear (but I have tried to give the authors a clue from my example of what could be an ideal title) I would want the authors to accept my feedback as mere suggestions that can help them to improve on the publication worthiness of their manuscript, starting from abstract to the conclusion (through the results and discussion). 

Author Response

Response to Reviewer 2

We want to thank the reviewer for the suggestions. It was a pleasure to try to integrate the manuscript following this suggestion because I think that a good review is important to improve the value of the manuscript. Thank you so much!

Very important: I want the authors to understand that my review has been critically constructive because their manuscript has some merit for publication. Their highly qualitative oriented methods fit a work of this sort, which needs the power of narration to argue its novelty. However, the novelty, in this case, is not yet clear (but I have tried to give the authors a clue from my example of what could be an ideal title) I would want the authors to accept my feedback as mere suggestions that can help them to improve on the publication worthiness of their manuscript, starting from abstract to the conclusion (through the results and discussion).

  • Title: the title of the paper, “Adopting a bottom-up and top-down participatory approach to plan and design local urban development” is simply a statement that is expressive of nothing interesting. It would have been more interesting and make for a good title if it was possible to get an idea of a unique reason for “Adopting a bottom-up and top-down participatory approach to plan and design local urban development”. Either “a bottom-up and top-down participatory approach” (or both) must be adopted “to plan and design local urban development” procedures. The title does not present a novel issue. An example (and this is based on my understanding of the paper) of a good title would have been something like “Combining bottom-up and top-down participatory approaches: Applications in the plan and design of an Urban University Centre”. With such a title, readers can easily see the innovation (how bottom-up and top-down approaches were combined) and see the application (planning and designing) and the object or case under investigation (Urban University Centre).

Response

Thank you very much for this comment. We rewrote the title following the suggestion of the reviewer

  • Abstract: The abstract does not directly relate to the reader precisely what the article is all about. It stresses the bottom-up and top-down processes carried out but do not address questions related to who carried out the processes, for what it was done, and why the approaches were adopted. I believe in the mind of the authors they believe they must have answered these questions, but their wording and phrasing of their narratives fail to carry the message across to me as a reader. For instance, in lines 21-22, the authors in expressing their objective (or critical motive of the research) stated that “This work intends to try to apply a planning process of urban spaces creating feedback between different stakeholders: community, planners, and decision-makers.” However, this does not relate any specific message.  Why not simply put it this way, “this work embraces a planning process that…”. The later is in the present while the former is in the future. It would have been more understandable if the manuscript was written mostly in the present (or combined with past where necessary) as it is reporting what has been done (in the past) in the present. This is a minor issue but needs to be addressed as it gets in the way of comprehending the article. Due to this kind of sentence wording (which are mostly not wrong) that are awkward, it is difficult to grasp the abstract fully. This abstract needs rewriting. The authors can improve the abstract by simply adapting specific sentences that reflect or answer the following questions: What is the background of the study? Why is the study necessary now? And why is the study being reported? How was the study done? What result or outcome did the study produce? What can we conclude from the outcome of the study? And how does it add to existing knowledge? If the authors can create one or two short and straightforward sentences that answer these questions, then the abstract will project-specific messages that touch on the relevance, objective, methodology, results and conclusion of the research.

Response

We rewrote the abstract following the suggestions of the reviewer.

  • Introduction: The introduction needs to provide a more robust background of the study, as well as justify the study with literature evidence. This is especially important since the authors did not introduce a literature-based section before the “materials and Methods” section. I will not go as far as suggesting that the authors include a literature review section. Still, for them to avoid this, they need to produce a robust and expanded introduction that justifies what has been done in a related field and what the missing pieces (or gaps) exist, and why this current study fills some gap or gaps going forward. These arguments should be backed with reliable literature.

Response

We implemented the Introduction trying to provide a background of the study In particular, we added one figure to represent the relation between institutional levels and urban scale in urban planning and design and we added a reliable literature.

  • Materials & Methods (Methodology): To have a better-structured manuscript, I will advise that the “study area” (section 2) be integrated with the “Materials & Methods” (section 3). What this will mean is that the “study area” is made a sub-section (preferably the first section) in the “Materials & Methods”. This will allow for directly putting the methodology into a case study context. In the current structure, the two are isolated from each other. More so, a case study is part of a methodology because it is the object under investigation. Without the case study, the entire research will lose context and scope, become scientifically meaningless. This is why I advise the authors to merge those two sections. Also, I think the authors should focus on describing the “study area” (that is, the Municipality of Lecce and location of the study area) in the “study area” sub-section, rather than describing the planning process. This should be followed by the “study design” sub-section, and….

Response

We reorganized this section moving the study area in the Materials and Methods and we split this section in two parts according to the comment of the reviewer.

  • Result & Discussion: The critical question which the research evoked but failed to answer is the question of, what is unique in the combination of “bottom-up” and “top-down” approaches as employed in the PUUC development? How different would the outcome have been if the two were not combined in the manner stated in the study? The answers to these questions are not easily decipherable from the study. Could it be that the answers are tucked away or hidden somewhere within the text? Maybe. But the authors can boldly state this by dedicating headings or sub-headings to make them visible in the discussion section. Another minor issue: What did the authors mean by putting lines 425-441 in italics? I didn’t get it?

Response

We changed the structure of the result and discussion to better explain the contribution of the bottom-up and top-down sections. We added new elements in discussion in agreement with other reviewers. We tried to better explain the limitation and implications of this work.

Lines 425-441 are considerations derived from the interviews. We remove the italics.

  • Conclusion: As the discussion failed to clearly identify the novelty of the study, it is difficult for me to assess the conclusion.

Response

We rewrote the conclusion section trying to explain the practical contributions and implications of this work and adding one figure.

Thank you for the valuable feedback on our manuscript. We appreciate the chance to revise and improve our work. The comments were all addressed in the updated manuscript and point-by-point below. We found the comments very helpful and believe that now the manuscript has been improved by taking them into account. 

Round 2

Reviewer 2 Report

First of all, I would like to thank the authors for the efforts they have put into improving this work. Somehow, I think there are areas that need (re)addressing.

TITLE: I am aware that in my previous review that I was critical of the title of the manuscript, and so must have driven the authors to want to improve it. However, the revised title, “A new urban share vision combining bottom-up and top-down participatory approaches in the socio-ecological system: Applications in the plan and design of an Urban University Center” is wordy and fails to relate a direct message to readers. Instead, it mixes many important messages up. The new title is muddled with about 10 variables –shared vision, bottom-up, top-down, participatory approaches, socio-ecological system, applications, plan, design…. These all make it difficult to quickly comprehend. Can the authors consider the following (and refine  as they deem suitable):

  • Bottom-up and top-down participatory approaches in socio-ecological system planning and designing: Experience from an Urban University Center
  • Bottom-up and top-down approaches: Application of a shared vision in socio-ecological system planning and designing an Urban University Center
  • Implementing a shared vision in socio-ecological system planning and designing of an Urban University Center
  • A bottom-up and top-down participatory approach to planning and designing local urban development: Evidence from an Urban University Center

These sort of titles will speak directly to the objective of the manuscript, which I believe is to showcase evidence of a multi-institutional level approach that is capable of embracing different visions and stakeholders' needs, as well as in combining bottom-up and top-down participation approaches.

PLACE FIGURES IN THEIR RELEVANT TEXTS IN THE MANUSCRIPT: Again it is important to put all figures between texts and explain. Placing Figures at the end of sections gives the impression of the figure not fully needed in the manuscript – because if the figure is necessary then it would have found a suitable place in the text and also been explained. Some Figures (e.g. Figures 1, 14, 13, 8…).

ABSTRACT/INTRODUCTION/METHODS: The abstract has been considerably improved. The introduction now provides a stronger background argument but Figure 1 needs to be enclosed within texts, directly below the text-lines where it has been mentioned in the manuscript. The authors have produced a more integrated methodological section.

DISCUSSION: The discussion captions my previous recommendations.

CONCLUSION: Please do not end an article with a Figure. I actually do not encourage having a figure in the conclusion, except it is a conclusion that is merged with discussion or recommendations. Why was this necessary? Especially as the authors did not put in an iota of effort in explaining the diagram. My suggestion would be that: if the authors want to include Figure 14 in the conclusion, then they need to recaption that section as “Conclusion and recommendation (or suggestions going forward”. They also need to put the Figure in between texts (immediately after it is mentioned. Then go ahead to specifically explain its message and relevance going forward.

LANGUAGE: Lines 526-529 says “This is important, because it allows us to have a vision of the development of the area that is not conditioned only by the cultural background of the decision-makers, but of those who use the territory, in an effort to meet the needs of everyday life (Figure 14).” Maybe the authors should do away with “us” in that sentence. For instance, saying it this way, “This is important, because it allows for a vision of the development of the area that is not conditioned only by the cultural background of the decision-makers, but of those who use the territory, in an effort to meet the needs of everyday life (Figure 14).”

Author Response

Response to Reviewer 2

We want to thank the reviewer for the suggestions and for the effort don to actively contribute to improving the manuscript

TITLE: I am aware that in my previous review that I was critical of the title of the manuscript, and so must have driven the authors to want to improve it. However, the revised title, “A new urban share vision combining bottom-up and top-down participatory approaches in the socio-ecological system: Applications in the plan and design of an Urban University Center” is wordy and fails to relate a direct message to readers. Instead, it mixes many important messages up. The new title is muddled with about 10 variables –shared vision, bottom-up, top-down, participatory approaches, socio-ecological system, applications, plan, design…. These all make it difficult to quickly comprehend. Can the authors consider the following (and refine as they deem suitable):…

Response

Done.

I chose “A bottom-up and top-down participatory approach to planning and designing local urban development: Evidence from an Urban University Center”

I think that it adapts well to the manuscript. Many thanks!

PLACE FIGURES IN THEIR RELEVANT TEXTS IN THE MANUSCRIPT: Again it is important to put all figures between texts and explain. Placing Figures at the end of sections gives the impression of the figure not fully needed in the manuscript – because if the figure is necessary then it would have found a suitable place in the text and also been explained. Some Figures (e.g. Figures 1, 14, 13, 8…).

Response

Done.

ABSTRACT/INTRODUCTION/METHODS: The abstract has been considerably improved. The introduction now provides a stronger background argument but Figure 1 needs to be enclosed within texts, directly below the text-lines where it has been mentioned in the manuscript. The authors have produced a more integrated methodological section.

Response

Thank you

DISCUSSION: The discussion captions my previous recommendations.

Response

Thank you

CONCLUSION: Please do not end an article with a Figure. I actually do not encourage having a figure in the conclusion, except it is a conclusion that is merged with discussion or recommendations. Why was this necessary? Especially as the authors did not put in an iota of effort in explaining the diagram. My suggestion would be that: if the authors want to include Figure 14 in the conclusion, then they need to recaption that section as “Conclusion and recommendation (or suggestions going forward”. They also need to put the Figure in between texts (immediately after it is mentioned. Then go ahead to specifically explain its message and relevance going forward

Response

Done

I explained better figure 14 and reorganized the conclusion section.

LANGUAGE: Lines 526-529 says “This is important, because it allows us to have a vision of the development of the area that is not conditioned only by the cultural background of the decision-makers, but of those who use the territory, in an effort to meet the needs of everyday life (Figure 14).” Maybe the authors should do away with “us” in that sentence. For instance, saying it this way, “This is important, because it allows for a vision of the development of the area that is not conditioned only by the cultural background of the decision-makers, but of those who use the territory, in an effort to meet the needs of everyday life (Figure 14).”

Response

Done
